# The Intake of Coffee Increases the Absorption of Aspirin in Mice by Modifying Gut Microbiome

**DOI:** 10.3390/pharmaceutics14040746

**Published:** 2022-03-30

**Authors:** Jeon-Kyung Kim, Min Sun Choi, Hye Hyun Yoo, Dong-Hyun Kim

**Affiliations:** 1Neurobiota Research Center, College of Pharmacy, Kyung Hee University, Seoul 02447, Korea; kim_jk0225@naver.com; 2Institute of New Drug Development, School of Pharmacy, Jeonbuk National University, Jeonju 54896, Korea; 3Institute of Pharmaceutical Science and Technology and College of Pharmacy, Hanyang University, Ansan 15588, Korea; mschoi@kirams.re.kr

**Keywords:** aspirin, salicylic acid, pharmacokinetic, gut microbiome, Mrp4

## Abstract

The absorption of orally administered aspirin into the blood was affected by gastrointestinal environmental factors such as gut pH, digestive enzymes, and microbiota. The intake of coffee affects the pharmacological effects of aspirin. Therefore, we examined the gut microbiota-mediated effect of coffee bean extract (CBE) intake on the pharmacokinetics of aspirin in mice. The intake of CBE modified the gut microbiota composition and their α- and β-diversities: It decreased the *Proteobacteria*, *Helicobacteriaceae*, and *Bacteroidaceae* populations in the fecal microbiota composition, while the S24-7_f (*Muribaculaceae*) and *Lactobacillaceae* populations increased. The fecal aspirin-hydrolyzing activities of humans and mice to salicylic acid were 0.045 ± 0.036 μmole/h/g and 0.032 ± 0.003 μmole/h/g, respectively. However, CBE treatment significantly suppressed the aspirin-hydrolyzing activity in mice. Furthermore, the area under the serum concentration–time curves (AUCs) of aspirin and salicylic acid were 0.265 ± 0.050 µg·h/mL and 16.224 ± 5.578 µg·h/mL in CBE-treated mice, respectively, and 0.248 ± 0.042 µg·h/mL and 10.756 ± 2.071 µg·h/mL in control mice, respectively. Moreover, CBE treatment suppressed the multidrug resistance protein 4 (Mrp4) expression in the intestines of mice, while the P-glycoprotein (P-gp), breast cancer resistance protein (BCRP) expression was not affected. Furthermore, the CBE-treated mouse fecal lysate suppressed Mrp4 expression in Caco-2 cells compared to that of vehicle-treated mice, while CBE treatment did not affect Mrp4 expression. Oral gavage of caffeine also suppressed the Mrp4 expression in the intestines of mice. These findings suggest that intake of coffee can increase the absorption of aspirin by modifying the gut microbiome.

## 1. Introduction

A drug interaction is a situation in which the pharmacokinetic, pharmacodynamic, and pharmacological parameters of a drug are affected by a substance such as drugs or diets [1,2]. Considering drug interactions, the pharmacokinetic alteration of drugs generally occurs because substances inhibit or induce transporters and enzymes involved in the absorption, metabolism, and excretion processes of drugs in the intestine and liver [3,4]. Concomitantly with these, many studies have demonstrated that gut microbiota are involved in drug interactions [5,6]. Regulators of gut microbiota such as diets and antibiotics significantly affect the pharmacokinetic parameters in vivo: Antibiotics treatment increases the absorption of amlodipine or aspirin in mice [7,8].

Aspirin, acetylsalicylic acid, is a commonly used medication for pain, rheumatic fever, and inflammation [9]. Its oral administration is rapidly absorbed from the gastrointestinal tract into the blood as mainly salicylic acid and aspirin, and aspirin is metabolized into salicylic acid during and after the absorption from the gastrointestinal tract into the blood [10,11,12]. Thus, the reduction in gastric acidity facilitates the absorption of aspirin due to the unionized form of aspirin [13,14]. The metabolite salicylic acid relieves pain, fever, and inflammation [15]. The absorbed salicylic acid is further metabolized in the liver by hepatic conjugating enzymes such as UDP-glucuronosyltransferase and secreted into the urine (>80%) or bile duct (<5%) [10,14,16,17,18]. Salicylate conjugates secreted into the intestine via the bile duct can be metabolized in salicylates by gut microbiota and reabsorbed into the blood [6,12]. Gastrointestinal environmental factors such as gut pH and microbiota, which degrade aspirin to ionized forms [13], may influence the reabsorption of salicylate conjugates into the blood. The suppression of gut microbiota by ampicillin increases the absorption of aspirin into the blood in rats orally treated with aspirin and its analgesic activity [12].

Caffeine significantly increased the analgesic activity of aspirin in patients with a sore throat and postoperative oral surgery pain [19,20]. Furthermore, the combination of aspirin with caffeine significantly increased the area under the plasma concentration–time curve (AUC) of salicylic acid in humans compared to those given aspirin alone [21,22]. Nevertheless, the effects of coffee on gut microbiota-involved pharmacokinetic parameters of aspirin have not been studied thoroughly. 

Therefore, the main objective of this work was to investigate the gut microbiota-mediated effect of coffee intake on the pharmacokinetic parameters of aspirin in mice.

## 2. Materials and Methods

### 2.1. Materials

Aspirin (acetylsalicylic acid) and salicylic acid were purchased from Sigma (St. Louis, MO, USA). A QIAamp Fast DNA stool Mini Kit was purchased from Qiagen (Hilden, Germany). An instant coffee extract (containing 13 mg/g caffeine) was purchased from Dongsuh Foods Cooperation (Seoul, Korea). The methanol, acetonitrile, and acetic acid used were HPLC-grade and purchased from J.T. Baker (Phillipsburg, NJ, USA).

### 2.2. Subjects

The subjects were 5 healthy male Korean persons (average, 40.00 ± 9.58 years; range, 23–51 years). Exclusion criteria included smoking and current medication, especially regular or current use of antibiotics and/or aspirin. The recruitment of subjects and the collection of their stools were approved by the Committee for the Care and Use of Clinical Study in the Medical School at Kyung Hee University (IRB No KHP-130304R1).

### 2.3. Animals

C57BL6 male mice (20–22 g, 8 weeks old) were supplied by the Orient Bio Animal Breeding Center (Seoul, Korea). All animals were housed in wire cages (3 mice per cage) under controlled conditions (temperature, 20–22 °C; humidity, 50 ± 10%; light/dark cycle, 12 h), fed standard laboratory chow, and allowed water ad libitum. All experiments were approved by the Committee for the Care and Use of Laboratory Animals in Kyung Hee University (IACC, KHPASP(SE)-17044) and performed in accordance with the NIH and Kyung Hee University Guides for Laboratory Animal Care and Use.

### 2.4. Fecalase Preparation

Human and mouse fecal specimens were prepared according to the method of Kim et al. [12]. Briefly, fresh feces (0.4 g) were collected in sterilized plastic cups, fully suspended in 3.6 mL of cold saline, and centrifuged at 500× *g* at 4 °C for 5 min. The supernatant was sonicated for 10 min and then centrifuged at 10,000× *g* for 20 min and used for the assay of enzyme activity.

### 2.5. Aspirin-Metabolzing Activity Assay

The reaction mixture (total volume of 2 mL) consisted of 0.2 mL of 0.25 mM aspirin, 1.6 mL of 0.1 M phosphate buffer (pH 7.0), and 0.2 mL of fecalase suspension [12]. The reaction mixture was incubated at 37 °C for 12 h. The reaction was stopped by the addition of 1 mL of MeOH, and centrifuged at 3000× *g* for 10 min. The amount of aspirin in the resulting supernatant was assayed by high-performance liquid chromatography (HPLC), as previously reported [12].

### 2.6. Fecal Enzyme Activity Assay

For the activity assay of fecal β-glucosidase, β-glucuronidase, and sulfatase, the reaction mixture (total volume of 0.2 mL), which contained 20 μL of 2.5 mmol/L p-nitrophenyl-β-D-glucopyranoside (Sigma-Aldrich) for β-glucosidase, 2.5 mmol/L p-nitrophenyl-β-D-glucuronide (Sigma-Aldrich) for β-glucuronidase, 2.5 mmol/L p-nitrophenyl palmitate (Sigma-Aldrich) for lipase, or 2.5 mmol/L p-nitrophenyl sulfate (Sigma-Aldrich) for sulfatase, 75 μL of 0.05 mol/L phosphate buffer (pH 7.0), and 20 μL of the fecalase, was incubated at 37 °C for 20 min and added 0.2 mL of 0.1 N NaOH, as previously reported [23]. Enzyme activities were indicated as the amount required to catalyze the formation of 1.0 nmole of p-nitrophenol per hour. Specific activity was defined in terms of μmol/h/g wet feces.

### 2.7. Analysis of Gut Microbiota Composition

The fresh feces (0.2 g) were collected in a sterilized tube on the 1st day (24 h) after the final treatment with CBE. The genomic DNA was extracted using a QIAamp DNA stool mini kit and amplified using barcoded primers, which targeted the V3 to V4 region of the bacterial 16S rRNA gene, as previously reported [12]. The sequencing was performed using a 454 GS FLX Titanium Sequencing System (Roche, Branford, CT, USA). Sequence reads were identified using the EzTaxon-e database (http://eztaxon-e.ezbiocloud.net/ (accessed on 15 October 2020)) on the basis of 16S rRNA sequence data. The number of sequences analyzed, the observed diversity richness (operational taxonomic units, OTUs), and the estimated OTU richness (ACE and Chao1) were calculated using the Mothur program and defined considering a cut-off value of 97% similarity with the 16S rRNA gene sequences. Sequenced reads were deposited in the NCBI’s short read archive under accession number PRJNA449459.

### 2.8. Quantitative Real-Time Polymerase Chain Reaction (qPCR)

CBE (1.7 g/kg/day) or caffeine (24 mg/kg/day) was administered via oral gavage once a day for 5 days. Aspirin was administered via oral gavage 24 h after the final administration of CBE. The small and large intestines were removed 24 h after treatment CBE or caffeine, washed with saline twice, homogenized in 50 mM Tris-HCl buffer (pH 7.4), and centrifuged (10,000× *g*, 4 °C, 30 min). The quantitative real-time polymerase chain reaction (qPCR) was performed.

The qPCR for the assay of the P-glycoprotein (P-gp), breast cancer resistance protein (BCRP), multidrug resistance protein 4 (Mrp4), glyceraldehyde 3-phosphate dehydrogenase (GAPDH), or β-actin expression was performed, as previously reported [24], utilizing the Qiagen thermal cycler, which used SYBR premix agents. Reverse transcription was performed with total RNA (2 μg) isolated from the small and large intestines. Thermal cycling was performed at 95 °C for 30 s, followed by 42 cycles of denaturation at 95 °C for 5 s and amplification at 72 °C for 30 s. The expression of genes was calculated relative to GAPDH. The primers for qPCR are described in Appendix A.

For the analysis of gut microbiota composition by qPCR, fresh mouse feces (0.2 g) were collected in a sterilized tube 24 h after the final treatment with coffee, and genomic DNA was extracted using a QIAamp DNA Stool Mini Kit. qPCR was performed with 100 ng genomic DNA with SYBR premix in a Takara thermal cycler [23]. Thermal cycling was performed at 95 °C for 30 s, followed by 42 cycles of denaturation at 95 °C for 5 s and amplification at 72 °C for 30 s. The expression of genes was calculated relative to 16S rDNA. The primers for qPCR are described in Appendix A.

### 2.9. Culture of Caco-2 Cells

For the analysis of drug transporter expression, Caco-2 cells were cultured in 12-well plates (1 × 10^7^ cells/well) and incubated with CBE (10 μg/mL), CBE-treated fecal suspension (10 μg/mL), or saline in DMEM containing 1% antibiotic-antimycotic and 10% FBS at 37 °C for 24 h and washed three times [25]. The cells were collected, mRNA was purified using the RNeasy Mini Kit (Qiagen, Hilden, Germany), cDNA was prepared, and qPCR was performed according to the same method with analysis of the sample separated from the colon.

For the preparation of the fecal suspension, the feces of each mouse treated with or without CBE (1.7 g) were collected in plastic tubes, suspended in 2.7 mL of cold saline, and centrifuged at 500× *g* and 4 °C for 5 min. The supernatant was sonicated for 2 min five times and then centrifuged at 10,000× *g* at 4 °C for 20 min. The resulting supernatant was used as a fecal suspension.

### 2.10. Pharmacokinetic Experiments 

CBE (1.7 g/kg/day) or the vehicle (saline) was administered via gavage once a day for 5 days. Aspirin (5 mg/kg/day) was administered via gavage 24 h after the final treatment with CBE. After oral administration of aspirin, whole blood was collected at 0.03, 0.5, 1, 2, 4, 6, 8, and 10 h by cannulating the carotid artery using a polyethylene tube (PE-50) [12]. The obtained blood samples were centrifuged and the separated plasma samples were carefully maintained at −20 °C until analysis.

### 2.11. Blood Sample Preparation and Calibration Curves

The deproteinized plasma samples were injected into the LC-MS/MS analysis system, containing 5 ng/mL phenacetin as the internal standard (IS), as previously reported [12]. Calibration curves were derived by preparing a calibration standard with a standard solution of 1250 ng/mL (aspirin or salicylic acid) in a concentration range of 250–10,000 ng/mL and plotting the ratio of the analyte concentration to the analyte peak area using least-squares linear regression. Calibration curve equations were y = 0.013x + 1.204 and y = 0.006x + 3.549 (correlation coefficients were more than 0.99), respectively.

### 2.12. LC-MS/MS Analysis

The LC/MS/MS system used for the analysis consisted of an electrospray ionization source (ESI), Xcalibur software for the analysis (ThermoFinigan, Somerset, NJ, USA), and an SP LC system equipped with a TSQ Quantum access TM triple quadrupole mass spectrometer (ThermoFinigan, USA). For chromatographic separation, Fortis C8 (Fortis, UK) was used for HPLC separation, and the analysis conditions are the same as previously reported [12].

### 2.13. Pharmacokinetic Analysis

The maximum plasma concentration (Cmax) and the time to reach Cmax (Tmax) for aspirin or salicylic acid were directly estimated from plasma concentration–time profiles and the area under the plasma the drug concentration–time curve (AUC) using WinNonlin Professional 3.1 software (Certara, Ann Arbor, MI) was calculated as previously reported [12].

### 2.14. Statistics

All data were expressed as the mean ± standard deviation (S.D.), and statistical significance was analyzed by one-way ANOVA, followed by Student’s t-test using SPSS version 24 (IBM Corporation, Armonk, NY, USA).

## 3. Results

### 3.1. Oral Intake of CBE Modified Gut Microbiota Composition in Mice

Diets and drugs affect gut microbiota composition in humans and animals. Therefore, to understand whether the intake of coffee could affect the gut microbiota, we administered coffee via oral gavage to mice and measured the gut microbiota composition using 16S rDNA sequencing analysis (Figure 1 and Appendix A). The intake of CBE significantly increased the α-diversity (OTU, Ace, and Chao1) and shifted the β-diversity (principal coordinate analysis [PCoA]). CBE intake decreased the *Proteobacteria* population at the phylum level, *Helicobacteriaceae* and *Bacteroidaceae* at the family level, while the S24-7_f (*Muribaculaceae*) and *Lactobacillaceae* populations increased. At the genus level, coffee treatment increased EF602759_g, HM124280_g, and EF406806_g populations belonging to *Bacteriodales*, while the *Bacteroides* and AY239469_g populations belonging to *Bacteriodales* decreased. At the species level, coffee treatment increased EU474208_s (), GQ157662_s, EF602759_g_uc, and EU457676_s populations belonging to *Bacteriodales*, while the EF603109_s, EF406459_s, EF406686_s, EF406536_s, EF604981_s, AB599946_s, and JRMQ_s populations belonging to *Bacteriodales* decreased.

Next, we analyzed the types of gut bacteria that settled in the mice treated with CBE using selective media *Enterobacteriaceae*-selective DHL, *Lactobacillaceae*-selective MRS, *Bifidobacteriaceae*-selective BL, m-*Enterococcus* (mEn), *Bacteroides*-selective BBE, and non-selective Columbia (CB) agar plates (Figure 2A). CBE treatment significantly increased the number of colonies grown in MRS and CB agar plates, while the number of colonies grown in BL, mEn, BBE, and DHL agar plates was not affected.

To confirm the modification of gut bacteria by CBE treatment, the gut microbiota composition was analyzed using qPCR (Figure 2B). CBE treatment weakly, but not significantly, decreased the *Deferribacteres* and *Actinobacteria* populations, while the *Firmicutes*, *Bacteroidetes*, γ,δ*-Proteobacteria*, and TM7 populations were not affected.

### 3.2. The Intake of Coffee Suppressed the Aspirin-Metabolic Activity of Gut Microbiota in Mice

Next, we examined the fecal aspirin-hydrolyzing activities of humans and mice to salicylic acid (Figure 3). The fecal activities were 0.045 ± 0.036 μmole/h/g and 0.032 ± 0.003 μmole/h/g, respectively. To understand whether coffee intake could influence gut microbiota-based aspirin-hydrolyzing activity, we assayed the fecal aspirin-hydrolyzing activity in mice treated with saline (vehicle) or CBE for 5 days. CBE treatment decreased the fecal aspirin-metabolizing activity in mice; these were 0.032 ± 0.003 μmole/h/g and 0.019 ± 0.007 μmole/h/g in mice treated with the vehicle and coffee, respectively. CBE treatment increased β-glucuronidase, β-galactosidase, lipase, and sulfatase activities. Of these, β-galactosidase activity alone was significantly increased by CBE treatment.

### 3.3. Pharmacokinetic Study of Aspirin in Mice Treated with or without CBE

To examine the effect of CBE intake on the pharmacokinetics of aspirin in vivo, we administered CBE via oral gavage to mice for 5 days, orally administered aspirin 24 h after the final gavage of aspirin, and measured the concentrations of aspirin and salicylic acid in the blood. The mean blood concentration–time profiles of aspirin and salicylic acid in control and coffee-treated mice are indicated in Figure 4. The Tmax, Cmax, and AUC values of aspirin were 0.080 ± 0.000 h, 0.817 ± 0.170 μg/mL, and 0.248 ± 0.042 μg·h/mL in the control mice, respectively. The Tmax, Cmax, and AUC values of salicylic acid were 0.468 ± 0.382 h, 5.250 ± 1.076 μg/mL, and 10.756 ± 2.071 μg·h/mL in the control mice, respectively. The Tmax, Cmax, and AUC values of aspirin were 0.080 ± 0.000 h, 0.922 ± 0.147 μg/mL, and 0.265 ± 0.050 μg·h/mL in the coffee-treated mice control group, respectively. The Tmax, Cmax, and AUC values of salicylic acid were 0.885 ± 0.325 h, 7.216 ± 1.600 μg/mL, and 16.224 ± 5.578 μg·h/mL in the coffee-treated mice, respectively.

### 3.4. The Intake of CBE Suppressed Mrp4 Expression in Mice

We also examined the effects of CBE on the expression of the intestinal transporters P-gp, BCRP, and Mrp4 in the intestines of mice (Figure 5). Oral gavage of CBE significantly suppressed the Mrp4 expression in the small and large intestines, while the P-gp and BCRP expression was not affected (Figure 5A). To confirm whether treatment with CBE or gut microbiota was associated with the expression of Mrp4, we examined the effect of CBE or gut microbiota on the expression of P-gp, BCRP, and Mrp4 in Caco-2 cells. However, treatment with CBE did not affect P-gp, BCRP, and Mrp4 expression (Figure 5B). Next, we treated the fecal microbiota of mice treated with the vehicle or CBE in Caco-2 cells and measured the expression of intestinal transporters. Treatment with the CBE-treated fecal microbiota suppressed the Mrp4, while the expression of P-gp and BCRP was not affected (Figure 5C).

Next, we investigated the effect of caffeine, a main constituent of CBE, on the expression of transporters in mice (Figure 6). Caffeine treatment significantly increased the Mrp4 expression in the small and large intestines, while P-gp and BCRP expression was not affected.

## 4. Discussion

Most orally administered drugs are absorbed intact from the intestine into the blood; they are generally resistant to gastrointestinal environments such as stomachic and bile juice and gut microbiota [3,5]. However, some can be transformed into active, non-active, toxic metabolites by digestive enzymes and gut microbiota [7,12,26]. Of these, gut microbiota can metabolize hydrophilic drugs and phytochemicals such as sulfasalazine and ginsenoside Rb1 into hydrophobic compounds such as 5-acetylsalylic acid and compound K, respectively [6,27]. Human gut microbiota reside in the small intestine and large intestine. They transform food and drug ingredients non-absorbed in the gastrointestinal tract [6,27]. The transformation (metabolism) of drugs by gut microbiota can significantly affect the absorption of drugs into the blood. The drug-metabolic activity of gut microbiota is comparable to that of the liver [4,5,6]. However, the drug-metabolic processes of drugs via gut microbiota are significantly different from those of the liver. The liver catalyzes the oxidative and conjugative reactions for drugs, generally resulting in producing hydrophilic metabolites, which excrete into the urine and bile duct. However, gut microbiota transformed the hydrophilic drugs into the hydrophobic metabolites via the reductive and hydrolytic reactions, resulting in increasing their absorptions into the blood [5,6]. The number and composition of gut microbiota were affected by gut environmental factors such as drugs and diets.

Therefore, the shift of gut microbiota composition by drugs and diets can affect the absorption of drugs into the blood in humans and animals [5,6,28]. Orally administered aspirin can be transformed into salicylic acid and 5-hydroxysalicylic acid by the fecal microbiota and liver enzymes of humans and rodents [12]. Therefore, when aspirin is orally treated in rodents and humans, aspirin and/or salicylic acid were detected in the blood. If orally administered aspirin is metabolized into salicylic acid in the intestine, the metabolite salicylic acid is not easily absorbed into the blood. However, when ampicillin is treated in mice before treatment with aspirin, the absorption of aspirin and salicylic acid is increased. These results suggest that the aspirin-hydrolyzing activity of gut microbiota may be suppressed by ampicillin treatment. In addition, Thithapandha reported that oral administration of aspirin with caffeine significantly increased the absorption of aspirin into the blood in humans due to the enhancement of gastric acid secretion by caffeine [29]. Coffee intake can modify gut microbiota and their metabolites in vivo. Nevertheless, studies on the effects of coffee intake on gut microbiota-mediated pharmacokinetics of aspirin remain unclear.

In the present study, we found that pretreatment with coffee for 5 days increased the AUC of aspirin and salicylic acid in mice orally treated with aspirin. Oral gavage of coffee significantly modified the gut microbiota composition in mice: It increased the α-diversity estimated operational taxonomic unit (OTU) richness and shifted the β-diversity (principal coordinate analysis (PCoA)). Furthermore, the coffee treatment decreased the gut microbiota-based aspirin-hydrolyzing activity. The combined treatment of aspirin with caffeine significantly increased the AUC of salicylic acid in humans compared to those given aspirin alone [19,20,21]. These results suggest that the intake of coffee can increase the absorption of salicylic acid and aspirin, which is quickly hydrolyzed to salicylic acid during and after absorption by inhibiting the aspirin-hydrolyzing activity of gut microbiota.

The combined treatment of aspirin with caffeine significantly increased the AUC of salicylic acid in humans compared to those given aspirin alone [21,22,29]. Thithapandha et al. suggested that the intake of caffeine could increase the absorption of orally administered aspirin into the blood in humans due to the enhancement of gastric acid secretion [29]. Oral administration of aspirin with caffeine significantly increases the analgesic activity in patients with a sore throat and postoperative oral surgery pain compared to those treated with aspirin alone [19,20]. However, we found that pretreatment with coffee or caffeine suppressed the expression of Mrp4, which is an efflux transporter of aspirin and salicylic acid [30,31], while P-gp and BCRP expression was not affected. The fecal microbiota of mice treated with coffee suppressed Mrp4 expression in Caco-2 cells compared to that treated with the vehicle. Coffee extract or caffeine treatment did not affect directly Mrp4 expression in Caco-2 cells. These results suggest that the intake of coffee and its component caffeine can increase the absorption of aspirin by inhibiting the expression of Mrp4, an efflux transporter.

In addition, Kim et al. reported that gut microbiota hydrolyzed aspirin to salicylic acid, and the suppression of its hydrolytic activity via treatment with antibiotics increase the absorption of aspirin and salicylic acid into the blood in rodents [12]. Treatment with CBE or caffeine suppressed the absorption of aspirin and salicylic acid into the blood. These results suggest that the intake of coffee may increase the absorption of salicylic acid and aspirin by inhibiting the aspirin-hydrolyzing activity of gut microbiota and Mrp4 expression. Furthermore, the intake of coffee may increase the absorption of Mrp4 transporter-dependent drugs into the blood. Further research on the effect of coffee intake on gut microbiota-mediated pharmacokinetics of Mrp4 transporter-dependent drugs and the difference in the gut microbiota-based aspirin-metabolizing activity between males and females and between antibiotics-treated humans treated with and without drugs, including antibiotics and aspirin, is necessary.

## 5. Conclusions

Based on these findings, frequent intake of coffee in patients taking aspirin can increase aspirin and/or salicylic acid by modifying the gut microbiome, leading to an increase in its pharmacological activity. The fluctuation of pharmacokinetic parameters via coffee intake may occur with other drugs.

## Figures and Tables

**Figure 1 pharmaceutics-14-00746-f001:**
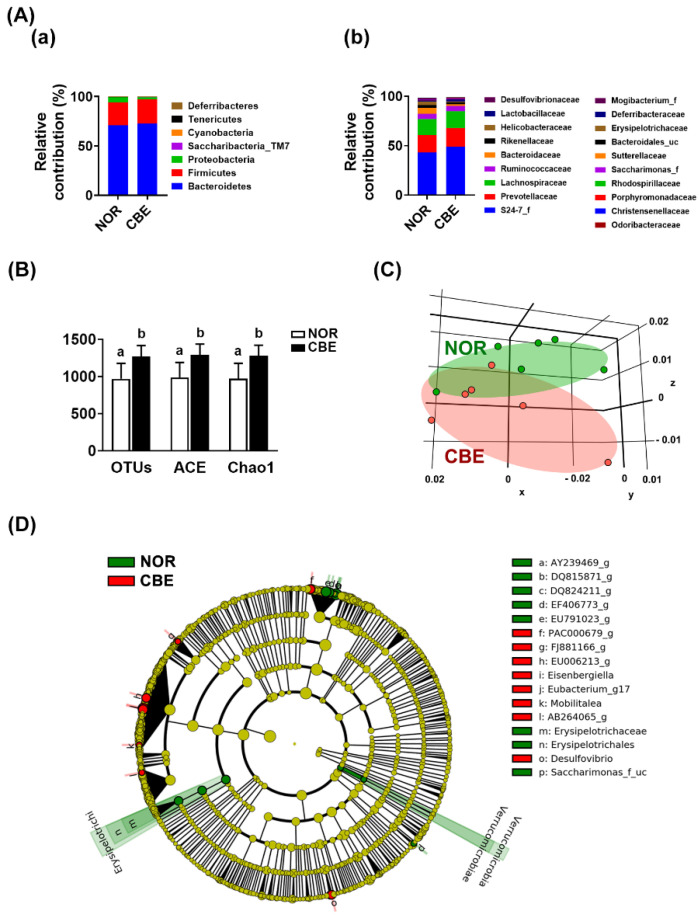
Effect of coffee bean extract (CBE) on the gut microbiota composition. (**A**) Effect on the gut microbiota composition. The relative composition at phylum (**a**) and family levels (**b**), assessed by pyrosequencing. (**B**) Effect on the α-diversity. (**C**) The principal coordinate analysis (PCoA) plot based on the pyrosequencing data. PCO1, principal component 1; PCO2, principal component 2. (**D**) Cladogram was generated via linear discriminant analysis effect size (LEfSE) analysis indicating significant differences in gut microbial abundances among normal control (NOR, green) and CBE-treated (CBE, red) groups. Data are shown as the mean ± S.D. (*n* = 6). Means with same letters are not significantly different (*p* < 0.05).

**Figure 2 pharmaceutics-14-00746-f002:**
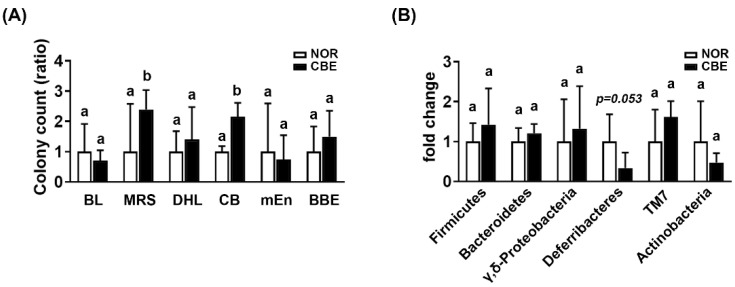
Effect of coffee bean extract (CBE) on the gut microbiota composition in mice treated with vehicle (NOR) or coffee bean extract (CBE). (**A**) Effect on the fecal microbiota composition, assessed by selective media (BL, MRS, DHL, CB, mEn, and BBE). (**B**) Effect on the fecal microbiota composition, assessed by qPCR. Data are shown as the mean ± S.D. (*n* = 6). Means with same letters are not significantly different (*p* < 0.05).

**Figure 3 pharmaceutics-14-00746-f003:**
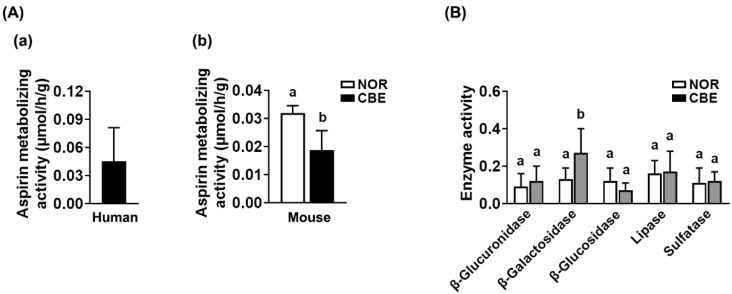
The enzyme activities of human and mouse fecal samples. (**A**) The fecal aspirin-metabolizing activities. (**a**) The fecal aspirin-metabolizing activities of humans. (**b**) The fecal aspirin-metabolizing activities fo mice treated with coffee bean extract (CBE) or vehicle (NOR) (**b**). (**B**) The β-glucuronidase, β-galactosidase, β-glucosidase, lipase, and sulfatase activities were analyzed in the feces of mice with CBE or vehicle. Data are shown as the mean ± S.D. (*n* = 6). Means with same letters are not significantly different (*p* < 0.05).

**Figure 4 pharmaceutics-14-00746-f004:**
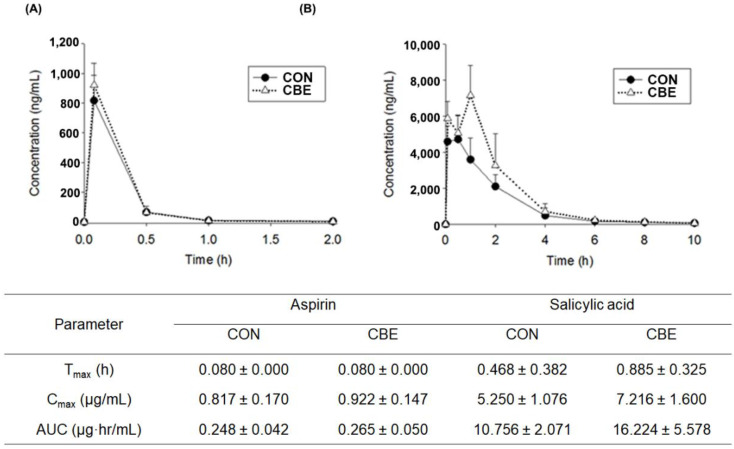
The plasma concentration of aspirin and salicylic acid. (**A**) The plasma concentration of aspirin in control (CON, treated with vehicle) and coffee bean extract (CBE)-treated mice. (**B**) The plasma concentration of salicylic acid in control (CON, treated with vehicle) and coffee bean extract (CBE)-treated mice. Aspirin (5 mg/kg) was orally or intravenously administered to mice for 3 days. Data shown are the mean ± S.D. (*n* = 6).

**Figure 5 pharmaceutics-14-00746-f005:**
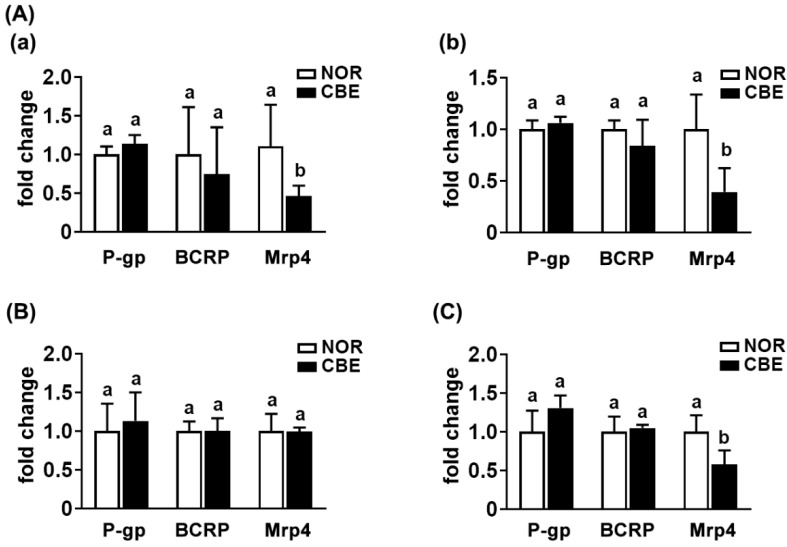
Effect of coffee bean extract (CBE) on the P-gp, BCRP, and Mrp4 expression in the intestines of mice. (**A**) Effect on the P-gp, BCRP, and Mrp4 expression. (**a**) Effect in the small intestine. (**b**) Effect in the large intestine. (**B**) Effects in Caco-2 cells treated with vehicle (NOR) or coffee bean extract (CBE). (**C**) Effect in Caco-2 cells treated with the fecal suspension of mice treated with vehicle (NOR) or coffee bean extract (CBE) on the transporter expression. The transporter expression was assessed by qPCR. Data are shown as the mean ± S.D. (*n* = 6). Means with same letters are not significantly different (*p* < 0.05).

**Figure 6 pharmaceutics-14-00746-f006:**
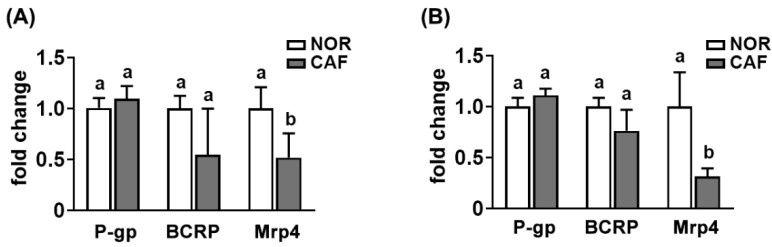
Effect of caffeine (CAF) on the P-gp, BCRP, and Mrp4 expression in the intestines of mice treated with vehicle (NOR) or caffeine (CAF). (**A**) Effect in the small intestine. (**B**) Effect in the large intestine. The transporter expression was assessed by qPCR. Data are shown as the mean ± S.D. (*n* = 6). Means with same letters are not significantly different (*p* < 0.05).

## Data Availability

Not applicable.

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
