# Peer review of "The Intake of Coffee Increases the Absorption of Aspirin in Mice by Modifying Gut Microbiome"

_pharmaceutics, 2022, doi:10.3390/pharmaceutics14040746_

Round 1

Reviewer 1 Report

The work presents very interesting results. The work is well written and presented, but needs a series of changes before being accepted.
 The format of all scientific names of microorganisms should be reviewed. For example, in the Abstract lines 16-18: the Proteobacteria, Helicobacteriaceae, and Bacteroidaceae populations in the fecal microbiota composition, while 17 the S24-7_f (Muribaculaceae) and Lactobacillaceae populations increased. Please must be in italics. Please correct it throughout the manuscript.

A homogeneous and unique criterion must be established to express the numerical data in the manuscript. Lines 18-19 use three decimal digits, followed by line 21 using two decimals. In other parts of the manuscript a decimal and up to four decimals are used. Please correct throughout the manuscript.
“Lines 18-19The aspirin-hydrolyzing activities of humans and mice to salicylic acid were 0.045 ± 0.036 μmole/h/g and 0.032 ± 0.003 μmole/h/g, respectively.”
“Line 21: and salicylic acid were 0.26 ± 0.05 ng h/mL and 16.22 ± 5.58 ng h/mL in CBE-treated mice, respectively, and 0.25 ± 0.04 ng h /mL and 10.76 ± 2.07 ng h/mL”
At the end of the introduction, the main objective of the work should be included. The main objective of this work was…
In the working group, 5 healthy men were used, why only 5 people if the study is not invasive? Why not work in another group with women? People with medical treatments have been excluded. There should be a group that received regular medical treatment with aspirin.
In the statistical analysis, indicate the software used for the analysis and better explain this section.
Figure 3 should be presented in another way A) a) and b) is confusing. It could be indicated in a title in the figure human group and mlice group.
In all the figures, the definitions of the abbreviations used must be reflected, even if they are already indicated in the manuscript. The figure by itself should be read without the need to go to other sections. Please review all figure legends in the manuscript. “Figure 3. The enzyme activities of human and mouse fecal samples. (A) Fecal aspirin-metabolizing activity in humans (a) and mice treated with and without CBE (b). (B) Fecal β-glucuronidase, β- galactosidase, β- glucosidase, lipase, and sulfatase activities. Data are shown as the mean ± S.D. (n = 6). Means with same letters are not significantly different (p < 0.05)”  

Best regards

Author Response

Reviewer #1

First of all, we greatly appreciate your and reviewers’ excellent suggestions. We revised our manuscript according to the suggestions of you and reviewers, and the revised sections in the manuscript are highlighted with yellow in the file of pharmaceutics-revision[1]-yellowhighlighted. The revised manuscript was grammatically edited by a native speaker.  

----------------------------------------------------------------------------------------

The work presents very interesting results. The work is well written and presented, but needs a series of changes before being accepted.

The format of all scientific names of microorganisms should be reviewed. For example, in the Abstract lines 16-18: the Proteobacteria, Helicobacteriaceae, and Bacteroidaceae populations in the fecal microbiota composition, while 17 the S24-7_f (Muribaculaceae) and Lactobacillaceae populations increased. Please must be in italics. Please correct it throughout the manuscript.

→ The scientific names of microorganisms have been checked throughout the manuscript and have been re-written in italic. (L16-18, L192-200, L210-211, L217-218)

A homogeneous and unique criterion must be established to express the numerical data in the manuscript. Lines 18-19 use three decimal digits, followed by line 21 using two decimals. In other parts of the manuscript a decimal and up to four decimals are used. Please correct throughout the manuscript.

“Lines 18-19 The aspirin-hydrolyzing activities of humans and mice to salicylic acid were 0.045 ± 0.036 μmole/h/g and 0.032 ± 0.003 μmole/h/g, respectively.”

“Line 21: and salicylic acid were 0.26 ± 0.05 ng h/mL and 16.22 ± 5.58 ng h/mL in CBE-treated mice, respectively, and 0.25 ± 0.04 ng h /mL and 10.76 ± 2.07 ng h/mL”

→ The numerical data have been expressed as three decimal digits throughout the manuscript. (L22-23; L168; L245-252; Figure 4).

At the end of the introduction, the main objective of the work should be included. The main objective of this work was…

→ The main objective of the work has been presented as the Reviewer suggested. (L64-65)

In the working group, 5 healthy men were used, why only 5 people if the study is not invasive? Why not work in another group with women? People with medical treatments have been excluded. There should be a group that received regular medical treatment with aspirin.

→ Thank you for your comment. We also agree with your suggestion. However, in the present study, we studied the effect of coffee on the gut microbiota-based aspirin-metabolizing activity. Generally the substrates of bacterial enzymes such as aspirin may be inducible in aspirin-metabolizing activity and antibiotics are significantly changeable. Therefore, in the present study, these groups were excluded. In further study, we hope to investigate the difference on the aspirin-metabolizing activity between male and female, between antibiotics-treated humans and healthy volunteers, between aspirin-treated group and aspirin-untreated group. (L76, L353-356)

In the statistical analysis, indicate the software used for the analysis and better explain this section.

→ The software used for the statistical analysis has been indicated (L183-184)

Figure 3 should be presented in another way A) a) and b) is confusing. It could be indicated in a title in the figure human group and mice group.

→ The labels for Figure 3 have been revised according to the Reviewer’s instruction.

In all the figures, the definitions of the abbreviations used must be reflected, even if they are already indicated in the manuscript. The figure by itself should be read without the need to go to other sections. Please review all figure legends in the manuscript. “Figure 3. The enzyme activities of human and mouse fecal samples. (A) Fecal aspirin-metabolizing activity in humans (a) and mice treated with and without CBE (b). (B) Fecal β-glucuronidase, β- galactosidase, β- glucosidase, lipase, and sulfatase activities. Data are shown as the mean ± S.D. (n = 6). Means with same letters are not significantly different (p < 0.05)” 

→ All the figure legends have been checked and the abbreviations have been defined.

Reviewer 2 Report

Kim et al., studied gut microbiota-mediated effect of coffee intake on the absorption of aspirin. They used cell culture, animal model and human samples in their study and suggested that the intake of coffee can increase the absorption of salicylic acid and aspirin. The study is interesting, however, it lacks human data and some important concentration-response experiments. Specific comments are below.

- No concentration response was done for the samples, drugs and CBE but used at different concentrations in different tests. Why and how did authors select those doses? Authors are encouraged to study the concentration-responses.

- Not enough human data are provided. Low n (5) value for human subjects, and higher age ranges among subjects (23-51 years old). The physiology of a young subject (~23 years old) may not be similar to older subject (~51).

- Did the authors consider any potential disease as an exclusion criterion? How did authors verify whether or not the subjects had any potential GI and/or digestive disorders and/or cardiovascular diseases especially for the older subjects?

- Similarly, did authors verify that the mice had no such diseases?

- How did authors verify whether or not the drugs and/or microbiota were affected by any potential disorders?

- What was the reason to use only male? Authors are encouraged to include both sexes of human subjects if possible, and both sexes of mice.

- It is confusing to understand whether some data were from human or mice. Specify the use of human samples clearly in the results as well as in the figure legends.

- Make sure the full forms of abbreviated terms are mentioned in the first place (e.g. multidrug resistance protein 4 (Mrp4).

Author Response

Reviewer #2

Dear Editor

First of all, we greatly appreciate your and reviewers’ excellent suggestions. We revised our manuscript according to the suggestions of you and reviewers, and the revised sections in the manuscript are highlighted with yellow in the file of pharmaceutics-revision[1]-yellowhighlighted. The revised manuscript was grammatically edited by a native speaker. 

---------------------------------------------------------------------------------------

Kim et al., studied gut microbiota-mediated effect of coffee intake on the absorption of aspirin. They used cell culture, animal model and human samples in their study and suggested that the intake of coffee can increase the absorption of salicylic acid and aspirin. The study is interesting, however, it lacks human data and some important concentration-response experiments. Specific comments are below.

- No concentration response was done for the samples, drugs and CBE but used at different concentrations in different tests. Why and how did authors select those doses? Authors are encouraged to study the concentration-responses.

→ Thank you for your comment. We pretreated in mice with coffee or caffeine. The dose of CBE (8.4 g/60kg) was decided on the base of three cups of coffee. Moreover, the content of caffeine was 13 mg/g in the present CBE. In the previous studies (including the report of Thithapandha et al, J Med Assoc Thai. 1989;72(10):562-6), to study the pharmacokinetic study of aspirin in human treated with caffeine, they treated 120 mg/person/day, which was based on the dose of APC [500 mg/day/person of aspirin + 500 mg/day/person of phenacetin + 130 mg/day/person]. The dose of aspirin (120 mg/day/person) was the content of CBE (approximately 2 mg/kg/day). The orally administered doses of CBE and caffeine (12-fold in mice, compared to those of humans) were decided, based on Km factor Guidance for the industry from the FDA. In further study, we are planning to perform concentration-responses.

(L124, L152, L157)

- Not enough human data are provided. Low n (5) value for human subjects, and higher age ranges among subjects (23-51 years old). The physiology of a young subject (~23 years old) may not be similar to older subject (~51).

→ Thank you for your comment. In the present study, we studied the effect of coffee on the gut microbiota-based aspirin-metabolizing activity. Like your suggestion, gut microbiota-based drug metabolizing activities significantly fluctuated by intrinsic and extrinsic factors, such as diets, hormones, antibiotics, etc. Therefore, in the present study, we did not consider these factors. In further study, we are planning to study the difference on the aspirin-metabolizing activity between male and female, between young generation and elderly, and between aspirin-treated group and aspirin-untreated group. (L353-356)

- Did the authors consider any potential disease as an exclusion criterion? How did authors verify whether or not the subjects had any potential GI and/or digestive disorders and/or cardiovascular diseases especially for the older subjects?

→ Thank you for your comment, We decide the exclusion criteria including gut microbiota dysbiosis-inducible factors such as antibiotics, smoking, and aspirin (which is aspirin-metabolizing activity-inducible) according to the previous report (Kim DH. Drug Metab Dispos. 2015;43(10):1581-9). (L353-356)

- Similarly, did authors verify that the mice had no such diseases?

→ No symptoms related to gastrointestinal problems were found throughout the entire experiment. Cardiovascular disease was not evaluated, but the possibility of having a disease is considered very slim because healthy mice used were provided by a certified laboratory animal company and maintained in the controlled condition.

- How did authors verify whether or not the drugs and/or microbiota were affected by any potential disorders?

→Thank you for your comment. We could not observe mortality, appetite loss, body weight loss, adverse reactions, or toxicologically relevant alterations. (L326-327)

- What was the reason to use only male? Authors are encouraged to include both sexes of human subjects if possible, and both sexes of mice.

→ Some papers reported sex differences in gut microbiota composition in mice (Gut Microbes. 2016. 313-322). Therefore, we agree with the Reviewer’s opinion on the necessity of experiments with both sexes of mice. However, in this study, we mainly focused on the interactions between coffee and aspirin and their association with the microbiome. Accordingly, to see the effects clearly, we conducted the study with single-gender groups. However, further studies should be followed to clarify the possible gender differences in CBE’s effects on aspirin pharmacokinetics. In further study, we are also planning to study the difference on the aspirin-metabolizing activity between male and female, between young generation and elderly, and between aspirin-treated group and aspirin-untreated group. (L353-356)

- It is confusing to understand whether some data were from human or mice. Specify the use of human samples clearly in the results as well as in the figure legends.

→ The labels have been revised according to the Reviewer’s instruction.

- Make sure the full forms of abbreviated terms are mentioned in the first place (e.g. multidrug resistance protein 4 (Mrp4).

→ The abbreviations used have been checked throughout the manuscript and their corresponding full names (including Mrp4) have been added if necessary. (L24)

Round 2

Reviewer 1 Report

I have no more comments

Author Response

Thank you for your comment.

Reviewer 2 Report

- Authors are encouraged to describe in details with appropriate references about how they selected the concentrations in the methods of the manuscript.

- Authors are encouraged to write a separate section “Limitations” in the manuscript and describe the limitations of the study (e.g., lack of concentration-response data, lack of human data, lack of age and sex difference studies, the factors that were not considered, other potential exclusion criteria that could be considered, the future study plans etc.).

- Authors are encouraged to make sure all the figure legends have n value mentioned.

Author Response

Thank you for your comment. We revised our manuscript according to your comments. The revised sections in the manuscript are highlighted with yellow in the PDF file of pharmaceutics-revision[2]-yellowhighlighted. I hope you will consider this paper as suitable for publication in Pharmaceutics.

----------------------------------------------------------------------

- Authors are encouraged to describe in details with appropriate references about how they selected the concentrations in the methods of the manuscript.

-->Thank you for your comment.

- Authors are encouraged to write a separate section “Limitations” in the manuscript and describe the limitations of the study (e.g., lack of concentration-response data, lack of human data, lack of age and sex difference studies, the factors that were not considered, other potential exclusion criteria that could be considered, the future study plans etc.).

-->Thank you for your comment.  We are planning to study the difference on the aspirin-metabolizing activity between male and female, between young and elderly groups, between antibiotics-treated humans and healthy volunteers,  and between aspirin-treated group and aspirin-untreated group

- Authors are encouraged to make sure all the figure legends have n value mentioned.

-->Thank you for your comment. We added it in all Figures.